# Nutritional Assessment in Outpatients with Heart Failure

**DOI:** 10.3390/nu16172853

**Published:** 2024-08-26

**Authors:** Regina López Guillén, María Argente Pla, Andrea Micó García, Ángela Dura de Miguel, Eva Gascó Santana, Silvia Martín Sanchis, Juan Francisco Merino Torres

**Affiliations:** 1Endocrinology and Nutrition Department, La Fe University and Polytechnic Hospital in Valencia, 46026 Valencia, Spain; regilop46@gmail.com (R.L.G.); mariaargentepla@gmail.com (M.A.P.); andrea.mico@hotmail.com (A.M.G.); angela.durademiguel@gmail.com (Á.D.d.M.); eva.gasco@hotmail.com (E.G.S.); silmarsan85@gmail.com (S.M.S.); 2Joint Research Unit on Endocrinology, Nutrition and Clinical Dietetics, Health Research Institute La Fe, 46026 Valencia, Spain; 3Department of Medicine, Faculty of Medicine, University of Valencia, 46010 Valencia, Spain

**Keywords:** heart failure, malnutrition, sarcopenia, GLIM, bioimpedance, nutritional ultrasound

## Abstract

Introduction: Heart failure (HF) is associated with significant alterations in body composition, including malnutrition due to insufficient intake, chronic inflammation and increased energy expenditure. Identifying the prevalence of malnutrition and the risk of sarcopenia in patients with HF is crucial to improve clinical outcomes. Material and methods: This cross-sectional, single-center, observational study involved 121 outpatients diagnosed with HF. Nutritional status was assessed using the Mini Nutritional Assessment (MNA), the Malnutrition Universal Screening Tool (MUST), and the Subjective Global Rating (SGA). Sarcopenia was screened using the SARC-F (Strength, Assistance in walking, Rise from a chair, Climb stairs, Falls) questionnaire and diagnosed based on the European Working Group in Older People (EWGSOP2) criteria and functionality with the Short Performance Battery (SPPB) test. Malnutrition was diagnosed according to the Global Leadership Initiative on Malnutrition (GLIM) criteria. Results: The study found that 10.7% had cardiac cachexia and 45.4% of deceased patients had been in this condition (*p* = 0.002). Moderate-to-high risk of malnutrition was identified in 37.1%, 23.9%, and 31.4% of patients according to the MNA, MUST, and SGA tests, respectively. According to the GLIM criteria, 56.2% of patients were malnourished. Additionally, 24.8% of patients had a high probability of sarcopenia, and 57.8% were not autonomous according to SPPB. Patients with less than 30% quadriceps muscle contraction were at a high risk of sarcopenia. Conclusions: There is a high prevalence of malnutrition among outpatients with HF, which is associated with worse prognosis, increased risk of sarcopenia, and greater frailty. These findings underscore the importance of early nutritional and functional assessments in this population to improve clinical outcomes.

## 1. Introduction

Heart failure (HF) is not simply a diagnosis, but a clinical syndrome consisting of cardinal symptoms such as dyspnea, edema, and fatigue, which may be accompanied by signs such as elevated jugular venous pressure, pulmonary crackles, and peripheral edema. It is caused by structural and/or functional impairment of the heart resulting in increased intracardiac pressure and/or inadequate ejection volume at rest and/or during exercise [1]. The prevalence in Europe is between 1 and 2% of adults [1]. It increases with age, with 1% being <55 years [1], 4.5% between 65 and 70 years [2], and greater than 10% >70 years [1]. With respect to sex, heart failure is more frequent in women [1].

The pathophysiology of malnutrition in HF is complex, and numerous factors are involved. On the one hand, the symptoms that occur in HF, such as dyspnea and fatigue, lead to increased energy expenditure, resulting in anorexia and impaired satiety. Likewise, intestinal edema and hypoperfusion and hepatic congestion can result in early satiety, nausea, anorexia, and malabsorption of nutrients. On the other hand, cardiac dysfunction induces a systemic response characterized by neurohormonal and inflammatory activation that are initially beneficial, but dysfunctional in the long term. Additionally, there is an imbalance between anabolic and catabolic pathways. The anabolic pathways are activated by testosterone and ghrelin. Appetite is stimulated by ghrelin, which is produced in the gastric fundus and induces the release of neuropeptide Y and agouti protein (AgPR), while satiety is mediated by leptin and adiponectin which are released in adipose tissue. Ghrelin induces the release of growth hormone, which leads to the secretion of insulin-like growth factor type 1 (IGF-1). This stimulates protein anabolism, which is key to muscle regeneration [1]. It has been seen that, in patients with HF, there is an excessive anorexigenic stimulation by leptin and a deficit of pro-orexigenic hormones such as ghrelin [3]. Testosterone is an anabolic steroid that binds to androgen receptors and promotes protein synthesis, and patients with cardiac cachexia have been found to frequently have low testosterone levels, leading to decreased muscle mass [4], and reduced functional capacity and survival [5]. Furthermore, in patients with heart failure and a high inflammatory state, three key protein degradation pathways are activated: the ubiquitin–proteasome system (UPS), autophagy, and apoptosis. The UPS is the main mechanism responsible for protein degradation [4]. Autophagy, a process that eliminates dysfunctional and unnecessary cellular components, is exacerbated by cardiac dysfunction, which triggers a systemic response mediated by neurohormonal and inflammatory activation, leading to the accumulation of autophagosomes in the muscle, prompting their degradation. Furthermore, in cardiac cachexia, increased proinflammatory cytokines intensify myocyte apoptosis through caspase activation [6]. A combination of inadequate protein intake, malabsorption, catabolic processes, inadequate anabolism, physical inactivity, and comorbidities can lead to malnutrition, sarcopenia, and cachexia [7,8].

Malnutrition is common in these patients, reaching 25–60% depending on the setting, whether outpatient or inpatient, and the screening method used [3,9,10]. It carries a poor prognosis, since the mortality range is 5–30% in one year and increases the risk of hospitalization by 18.9–65% [9].

Cachexia is defined as weight loss ≥5% in the absence of edema in the last 12 months (or BMI < 20 kg/m^2^) in patients with chronic disease and at least three laboratory criteria: anorexia, decreased muscle strength, low fat-free mass index, fatigue, and altered biomarkers (increased *C*-reactive protein, IL-6, hemoglobin < 12 g/dL or albumin < 3.2 g/dL) [11]. The prevalence of cardiac cachexia varies between 8 and 42% depending on the definition of cachexia used and the study population [12]. It has been found that, using the definition of the Cachexia Consensus Conference, its prevalence is around 5–10% [13]. It is more frequent in advanced HF and reduced LVEF. The prognosis is poor, with mortality reaching 50% at 18 months [14].

Moreover, the prevalence of sarcopenia in HF is between 25 and 60% depending on the diagnostic criteria used [15,16,17]. Increased catabolic stress in the skeletal muscle of CHF patients results in exercise intolerance, respiratory inefficiency, chronotropic incompetence, and insulin resistance, contributing to the catabolic state that results in limiting the functional status of patients.

Prevention and detection of malnutrition in HF patients is essential to improve their prognosis. In this regard, regardless of the method of nutritional assessment used, several studies have shown that malnutrition is an independent predictor of mortality. Lv, Ru et al. compiled a meta-analysis [9] that included 31 studies (12,537 patients with HF), showing that mortality in malnourished patients was twice as high as in normo-nourished patients, regardless of age. Jiang et al. conducted a study [18] that enrolled 209 patients hospitalized in the Intensive Care Unit with severe cardiovascular disease; 92.34% were at risk of malnutrition and 18.18% were at high risk of malnutrition. In their sample, the higher the nutritional risk, the higher the incidence of cardiovascular events and the higher the mortality. Guerra Sánchez et al. [19] included 76 patients hospitalized with HF, 22.4% of whom were malnourished and 77.6% were at risk of malnutrition according to SGA. In the group that underwent nutritional intervention, analytical parameters improved (increase in hemoglobin, total protein, and albumin) and results improved in the 6 min walk test. Evaluating the nutritional status of these patients is crucial as it allows for guiding the nutritional approach necessary to prevent or reverse malnutrition, contributing to an improved prognosis.

As we have seen, malnutrition in patients with HF is a poor prognostic factor that compromises the patient’s health and can lead to death, as in the case of cardiac cachexia. Therefore, the assessment of nutritional status in patients with HF is essential and will allow us to guide the nutritional approach to reverse the situation that compromises the patient’s life. Before being able to carry out nutritional intervention studies on these patients, the reality of the nutritional status of outpatients with HF must be known; hence, the main objective of this study was to determine the prevalence of malnutrition and risk of malnutrition in outpatients with HF, using the GLIM criteria and morphofunctional assessment [20]. As a secondary objective, the relationship between different nutritional screening tests and the GLIM criteria in HF patients was tested.

## 2. Materials and Methods

### 2.1. Study Design

This is a cross-sectional, single-center, observational study of 121 patients diagnosed with heart failure and under follow-up in outpatient cardiology clinics at the Hospital Universitario y Politécnico La Fe in Valencia. Data were collected between 1 February 2022 and 31 October 2023. All patients were diagnosed with HF according to the guidelines of the European Society of Cardiology [1]. Inclusion criteria included age between 18 and 80 years and being on active treatment for HF. Pregnant women and patients on palliative treatment were excluded. The entire sample was on a low-salt diet according to the recommendations of their cardiology physicians. Patients were randomly selected from the monographic consultation of heart failure in cardiology outpatient clinics. The study was approved by the Ethics Committee of the Instituto de Investigación Sanitaria La Fe and informed consent was obtained from each patient to use their data anonymously.

### 2.2. Nutritional Screening and Assessment

Nutritional screening was performed with the Spanish version of the MNA test [21] (Mini Nutritional Assessment), only in patients over 65 years of age. The MNA test was provided by Société des Produits Nestlé SA 1994, Revision 2009, Vevey, Switzerland, holders of the registered trademark, who hold the copyright (http://www.mna-elderly.com/, accessed on 20 May 2024).

In addition, nutritional screening with the MUST test [22] (Malnutrition Universal Screening Tool) and the SGA test [23] (Subjective Global Rating) was carried out on the entire sample. The MUST test was provided by members of the Malnutrition Action Group, Standing Committee of the British Association for Parenteral and Enteral Nutrition (BAPEN), available at (https://www.bapen.org.uk/, accessed on 20 May 2024). The SGA test was provided by the American Society for Parenteral and Enteral Nutrition (ASPEN) in the *Journal of Parenteral and Enteral Nutrition*, Detsky et al. 1987 [22], available at (https://nutricionemocional.es, accessed on 20 May 2024). The score obtained and the category in which the patients were classified as normo-nourished, at risk of malnutrition, or malnourished were recorded. In addition, anamnesis was performed, including nutritional history: usual weight, weight loss perceived by the patient (if any), and over what period of time. The data used to complete the tests were obtained through physical examination and a personal interview conducted by a trained nutritionist.

We evaluated the GLIM criteria. For the first step of these criteria, involving nutritional assessment, we used the MNA, MUST, and SGA tests. The second step involved the diagnosis of malnutrition based on three phenotypic and two etiological components. The phenotypic criteria were (a) unintentional weight loss >5% in the last 6 months or >10% beyond 6 months, (b) body mass index (BMI) <18.5 kg/m^2^ for those younger than 70 years or <20 kg/m^2^ for those older than 70 years, and (c) reduced muscle mass. The GLIM criteria suggest that reduced muscle mass should be diagnosed based on an assessment of body composition (e.g., bioimpedance analysis, computed tomography, or dual-energy X-ray absorptiometry). Muscle mass was determined by classical anthropometric measurements such as brachial circumference (BC), arm muscle circumference (AMP), tricipital crease (TP), and calf circumference (CC); the result is shown in percentiles according to standardized tables. Grip strength was measured three times bilaterally with the Jamar^®^ dynamometer (manufacturer JLW^®^, Chicago, IL, USA). For this test, the individual was positioned with the shoulder adducted at zero degrees of rotation, the elbow flexed at 90 degrees, and the wrist in neutral position. The average of the three measurements was used for analysis. Values below the fifth percentile of the Spanish normative reference data were considered a reduction in muscle strength. Similarly, muscle mass was measured by bioimpedancemetry and nutritional ultrasound^®^, as will be explained below. The etiological components of the GLIM criteria are (a) reduced intake (<50% of energy requirements for >1 week or any reduction >2 weeks) and (b) chronic inflammatory pathology. Malnutrition was diagnosed with the presence of at least one phenotypic and one etiologic criterion.

### 2.3. Body Composition Assessment

Body composition was evaluated by vector bioimpedance impedance tomography (BIVA) and nutritional ultrasound^®^. For BIVA, the NUTRILAB^®^ bioimpedance meter, distal tetrapolar, monofrequency, and vectorial at 50 kHz (manufacturer AKERN^®^, Pontassieve, Italy), was used. To begin with, the patient’s sex, date of birth, weight, and height were entered. The patient was then placed in the supine position and a BIATRODES™ electrode was placed on the metacarpal line (between the second and third fingers) and on the metatarsal line (between the second and third fingers) of the dominant hemisphere.

As for ultrasound, the U PROBE-L6C^®^ (manufacturer Léleman^®^, Valencia, Spain) (linear 7.5–10 KHz) ultrasound scanner was used, which allowed a depth of up to 100 mm. It was performed with the patient in supine decubitus in a relaxed position with the determination of the measurement locations in the leg and abdomen. The measurement location of the leg was the lower third between the anterior superior iliac spine and the upper edge of the patella. In the cross-section, the X and Y axes were measured, which correspond to the linear measurement of the distance between the muscular limits of the anterior rectus quadriceps muscle: lateral (X axis) and antero-posterior (Y axis). The muscle area was assessed by manual tracing around the edge of the muscle aponeurosis and the measurement of adipose tissue thickness was established as the linear distance between the epidermis and the aponeurosis of the quadriceps rectus anterior muscle. The second component was the evaluation of fat at the level of the abdominal wall. The location of the measurement point was established at the midpoint between the xiphoid appendix and the umbilicus in the midline. In the cross-section, the anatomical structures that were visualized were ordered from the most superficial layer corresponding to the epidermis, followed by the superficial and deep adipose tissue layer, and then the two anterior rectus abdominis muscles that meet in the central part of the linea alba were identified. Finally, the preperitoneal fat layer that lies between the lower border of the linea alba and the line of the parietal peritoneum was located.

### 2.4. Assessment of Sarcopenia and Functionality

Sarcopenia screening was performed using the validated Spanish version of the SARC-F test (Strength, Assistance in walking, Rise from a chair, Climb stairs, Falls) to determine whether patients have a low (<4 points) or high probability of sarcopenia (≥4 points). The SARC-F test was provided by Parra-Rodriguez et al. [24] in *The Journal of Post-Acute and Long-Term Care Medicine*, available at (https://www.jamda.com/, accessed on 20 May 2024). The diagnosis of sarcopenia was made using the EWGSOP2 (European Working Group on Sarcopenia in Older People) criteria, by which patients are divided into groups: (a) probable sarcopenia: low muscle strength measured by dynamometry (<27 kg in men and <16 kg in women) or the chair rise test (>15 s for five lifts); (b) diagnosis of sarcopenia: if there is coexistence of low muscle strength with the previously described criteria and low muscle quantity or quality defined as appendicular skeletal muscle mass (ASM) <20 kg in males and <15 kg in females or appendicular skeletal muscle mass between size^2^ (ASMI) <7 kg/m^2^ in males and <5.5 kg/m^2^ in females; and (c) severe sarcopenia if, in addition to low strength and low muscle quantity/quality, there is low physical performance. For the assessment of physical performance, the SPPB (Short Performance Battery) test was used, which consists of three tests: balance (feet together, semi tandem, and tandem), walking speed (over 4 m), and the chair rising test. With it, patients are divided into disabled (<3 points), frail (3–8 points), pre-fragile (8–10 points), or autonomous (>12 points). For the diagnosis of severe sarcopenia, SPPB test ≤ 8 points was used.

### 2.5. Other Clinical Parameters

The severity of HF was based on the NYHA (New York Heart Association) scale. Variables related to comorbidities such as hypertension (HT), dyslipidemia (LD), diabetes (DM), and pharmacological treatment were also gathered.

### 2.6. Statistical Analysis

Data were processed with the SPSS version 25.0 statistical package. Continuous variables are described as mean and standard deviation (SD) and median and interquartile range (IQR). Categorical variables are described as proportions (%). Qualitative variables were compared by Chi-square test and quantitative variables by using the Student’s *t*-test/ANOVA after checking the normality of the distribution using the Kolmogorov–Smirnov test. A value of *p* < 0.05 was considered statistically significant.

## 3. Results

### 3.1. Patient Characteristics and Nutritional Assessment

Age was 66 (57.7–73.5) years; 79 (65.3%) were male and, of the total sample, 11 (9.1%) died during data collection, of which 6 were males and 5 females. The duration of HF was 6 (3–11) years. Further, 42 (34.7%) had reduced LVEF, 23 (19%) moderately reduced, and 52 (43%) preserved. Regarding the NYHA scale, 52 (43%) were in NYHA I, 40 (33.1%) in NYHA II, 26 (21.5%) in NYHA III, and 3 (2.5%) in NYHA IV. Cardiovascular risk factors and BMI in all patients as well as in men and women are shown in Table 1 and Table 2.

In those with DM type 2, the mean of glycosylated hemoglobin (HbA1c) was 6.8 ± 0.9%. Regarding the usual pharmacological treatment, 22 (18.1%) received standard HF treatment consisting of beta-blockers, mineral corticoid receptor antagonist, iSGLT2, and sacubitril–valsartan.

In reference to the screening tests, MNA was performed on 70 patients (those > 65 years), while MUST and SGA were performed on the entire sample. Of the entire sample, 37.1%, 23.9%, and 31.4% were at moderate–high risk of malnutrition according to the MNA, MUST, and SGA, respectively.

The percentage of patients at moderate/high risk of malnutrition according to screening tests in deceased patients and with symptoms at low exertion/at rest can be seen in Figure 1.

According to the GLIM criteria for the diagnosis of malnutrition, the results in all patients and the distribution by sex are represented in Table 3. Table 4 shows the comparison between malnutrition measured by the MNA, MUST, and SGA and that detected by the GLIM criteria.

The percentage of weight loss was 1.8 ± 4.2%, with a minimum of 0% and a maximum of 23.7%; 15.7% had weight loss ≥ 5%. The time to weight loss was 4 (IQR 2–6) months. Among patients who had died, weight loss was 6.9 ± 8% vs. 1.3 ± 3.3% of those who were still alive (*p* = 0.000). Of the total, 13 (10.7%) had cardiac cachexia: 6 males and 7 females. Among the deceased patients, 45.4% had pre-cardiac cachexia (*p* = 0.002).

### 3.2. Sarcopenia and Functional Assessment

Regarding sarcopenia screening tests, the SARC-F sarcopenia screening test revealed that 91 (75.2%) had a low probability of sarcopenia and 30 (24.8%) had a high probability of sarcopenia. Using the EWGSOP2 criteria for sarcopenia, 81 (66.9%) had no sarcopenia, 25 (20.7%) had confirmed sarcopenia, and 15 (12.4%) had severe sarcopenia. Finally, regarding functional assessment (SPPB), 51 (42.1%) patients were autonomous, 49 (40.5%) pre-fragile, 16 (13.2%) frail, and 5 (4.1%) disabled. Table 5 shows the diagnosis of malnutrition and sarcopenia based on the NYHA scale.

On the other hand, in reference to grip strength measured by dynamometry, 45 men (56.9%) were below the 10th percentile according to age- and sex-standardized tables, while 42 women (61.9%) were also <p10. In males, 7.6%, 22.8%, and 2.5% were <p10 in the measurements of arm circumference, arm muscle circumference, and tricipital fold, respectively. As for women, 7.1% had brachial circumference <p10, 19% arm muscle circumference <p10, and 14.3% tricipital crease <p10. Regarding calf circumference, 10.1% of men and 16.7% of women had measurements ≤31 cm.

### 3.3. Evaluation of Body Composition

Bioimpedancemetry and nutritional ultrasound^®^ were used as complementary explorations. The measurements obtained by vector bioimpedancemetry in men are shown in Table 6, and in women in Table 7.

Regarding nutritional ultrasound^®^, the values obtained in the abdominal wall are shown in Table 8, and in the anterior rectus abdominis muscle in Table 9.

In men, there was a positive correlation (0.415) between PA and muscle area, such that when muscle area increased, PA increased (*p* < 0.001). In women, muscle area presented a correlation of 0.345 with cell mass (BCM) measured by BIA (*p* = 0.084).

We also calculated the percentage of contraction of the quadriceps muscle, understood as the increase in the Y axis in contraction with respect to the Y axis at rest. Males presented a contraction of 10.1 (−9.7–27)% and females 0 (−12.1–14.2)%. Similarly, we tested the contraction of the quadriceps muscle using 15%, 20%, and 30% as cut-off points. Table 10 shows the different cut-off points in relation to the risk of sarcopenia measured with the SARC-F test in the study population.

## 4. Discussion

Malnutrition is frequent in patients with HF. In our study, the prevalence of cardiac cachexia is 10.7%, similar to that reported by Christensen et al. [25] in which it is 10.5%; however, in our case, the diagnosis of cachexia was made using the 2006 cachexia consensus criteria, while in their study, patients with cardiac cachexia were diagnosed if there was a weight loss >10%. Similar to our study, Carson et al. [26] used the 2006 cachexia consensus criteria, finding a prevalence of 15%. This figure is higher than that observed in our study, which may be due to the fact that 10.5% of their sample were oncology patients and the patients were in NYHA III and IV, i.e., with more advanced HF. Cardiac cachexia worsens the prognosis, as it increases mortality by 5–30% within a year and the risk of hospitalization by 18.9–65% [9]. In our series, 45.4% of the deceased patients suffered pre-cardiac cachexia. These results are consistent with other studies [26], which show a 50% mortality in patients with cachexia after an 18-month follow-up, compared with 17% in those without cachexia.

For screening for malnutrition, the MNA test, MUST, and SGA were used. Several studies support their use in patients with HF. In one of them, Guerra-Sánchez et al. [27] compared the MNA, NRS-2002 (Nutritional Risk Screening 2002), CONUT (Nutritional Control), MUST, and the Cardona method with respect to SGA for screening for malnutrition in HF, concluding that the MNA and NRS-2002 were the most valid. In our study, NRS-2002 was not used since it is intended for hospitalized patients. Another study, also by Guerra-Sánchez et al. [19], used the MNA test, SGA, and/or weight loss without edema >5% for the diagnosis of malnutrition, finding 77.6% and 22.4% of patients at risk of malnutrition and malnourished, respectively, according to the SGA. Similarly, Santos Barbosa et al. [28] found malnutrition or risk of malnutrition in 75.3% of patients in their sample, using the SGA and MNA, and associated malnutrition with longer hospital stay. In another study, Duarte et al. [29] evaluated the association between grip strength, thumb adductor muscle thickness, and nutritional status, finding that 42.4% had moderate–high risk of malnutrition according to the SGA and association between nutritional status and HF severity measured with any of the above. These values are higher than those of our sample, where 37.1% and 31.4% present moderate–high risk of malnutrition according to the MNA and SGA, respectively. The difference may be due to the fact that the population of the aforementioned studies was older and had a higher percentage of patients in NYHA III–IV.

The GLIM criteria were used for the diagnosis of malnutrition. There are few studies evaluating the concordance between screening methods and the GLIM criteria in HF. One of them [30] found concordance between the GNRI (Geriatrics Nutritional Risk Index) test and the GLIM criteria in patients older than 65 years who were hospitalized for HF. Another study [31] found concordance between the MNA test and the GLIM criteria, but only the MNA test was an independent predictor of mortality and hospital admissions. In our study, we observed that the MNA test, MUST, and SGA were useful for predicting malnutrition, since patients who are classified as normo-nourished are not malnourished according to the GLIM criteria, and those at high risk of malnutrition are mostly malnourished according to GLIM. In our sample, 43.8% were normo-nourished, 45.5% were moderately malnourished, and 10.7% were severely malnourished, according to the GLIM criteria, with no differences by sex. These results differ from the literature, which reports a lower percentage of malnutrition according to the GLIM. Kootaka et al. [32] included patients with cardiovascular disease, of whom 45.9% had HF, reporting 18.9% malnutrition. Ito et al. [33] found 18.7% malnutrition in patients with non-ischemic dilated cardiomyopathy. In the same sense, Joaquin et al. [31] reported 19.9% moderate and 5.3% severe malnutrition in patients similar to our sample (ambulatory, in NYHA II, and age 69 years). This discordance may be due to the fact that our study includes patients with advanced HF, while in their study no patient was in NYHA IV.

Weight loss worsens prognosis in patients with HF; in fact, a criterion for presenting cardiac cachexia is weight loss ≥5% in the absence of edema, as established at the Cachexia Consensus Conference [13]. Our sample presents weight loss of 1.8 ± 4.2%, and 15.7% had weight loss ≥5%. Similar weight loss has been seen in other series, such as in the study by Seko et al. [34], in which 13.1% had weight loss ≥5% at the 6-month visit. In another study by Okuhara et al. [35], in which patients with CHF and NYHA II–III were evaluated, they saw weight loss ≥5% in 11.2% at the annual visit, slightly lower than our study, probably because it did not include patients in NYHA IV. On the other hand, other series have seen greater weight loss, as in the study by Trullas et al. [36], in which 20.8% had weight loss ≥5%. This increase may be due to the fact that the age of the population included was 78 years and 43.1% of the patients were in NYHA III–IV.

In patients with HF, it is essential to perform a morphofunctional assessment that includes body composition techniques such as bioimpedancemetry and nutritional ultrasound^®^, since the classic parameters can mask malnutrition. In an article by García Almeida et al. [20], a comparison is made between classic parameters (weight loss, BMI, folds, circumferences, albumin levels, lymphocytes, cholesterol, and intake) vs. advanced parameters (bioimpedanciometry and nutritional ultrasound^®^) in clinical nutrition, and the importance of assessing body composition in these patients is highlighted. In our case, the classic parameters are very limited as they can be masked by edema, water overload, and inflammation. In our series, 15.7% had weight loss ≥5% and 5.7% had weight loss ≥10%. Further, 6.6% had low BMI (<20 kg/m^2^ in those younger than 70 years or <22 kg/m^2^ in those older than 70 years) [37]. Regarding anthropometric parameters, 7.6% of men and 7.1% of women had an arm circumference <p10, 22.8% of men and 19% of women had arm muscle circumference <p10, and 2.5% of men and 14.3% of women had atricipital fold <p10. Therefore, if we had assessed malnutrition only with the classical parameters, it would not have reached 25% of our sample. When applying the GLIM criteria, which include morphofunctional assessment in their phenotypic criteria, 45.4% present moderate malnutrition and 10.7% severe malnutrition.

In our sample, the PA was 5.1 ± 1.1° in men and 4.7 ± 0.8° in women. Scicchitano et al. [38] found a PA of 5 ± 1.3° in patients with HF, which was lower in women. These values are lower than those found in the healthy population; in a study [39] conducted in a healthy population, PA values of 7.5 ± 1.1° were found in men and 6.5 ± 1.1° in women. PA may be influenced by the state of overhydration of these patients, so it is also necessary to evaluate BCM. In our sample, the BCM in men was 31.6 ± 7 kg vs. 21.8 ± 4.9 kg in women, indicating a difference between sexes. In addition, we observed a positive correlation between PA and BCM, so that the higher the BCM, the higher the PA. There are no published studies that analyze the value of BCM in patients with HF but, as in our sample, a positive correlation between PA and BCM was observed in healthy patients [40] and in patients with chronic kidney disease [41]. This opens the field for future research on BCM in HF given the hydration status of these patients.

Regarding nutritional ultrasound, in this study, it was applied to assess muscle mass and adipose tissue as part of the morphofunctional evaluation of patients with heart failure. It is important to note that while nutritional ultrasound is an emerging tool with significant potential for enhancing the accuracy of body composition assessments, it is not yet widely adopted in all clinical nutrition services. The technique is currently more common in research settings and specialized clinics, and the establishment of standardized reference values is still in progress. Despite these limitations, the inclusion of ultrasound in this study provided valuable insights into the muscle and adipose tissue characteristics of our patient population, which are critical for the accurate diagnosis of sarcopenia and malnutrition. Roman et al. [42] analyzed cut-off points for the anterior rectus quadriceps muscle in relation to sarcopenia measured by EWGSOP2 criteria. In our sample, 66.9% did not have sarcopenia according to EWGSOP2 but were at risk due to their chronic disease. The cut-off points in patients at risk of sarcopenia were 3.48 cm^2^ in males and 2.97 cm^2^ in females, and our population falls below these cut-off points. This could be due to the fact that they present a chronic disease that conditions an inflammatory state and a higher average age, 66 years, than in other studies. In line with our findings, a study by Fuentes-Abolafio et al. [43] in patients aged 80.7 years with HF and preserved LVEF observed an area of the anterior rectus quadriceps muscle in men of 2 cm^2^ and 1.98 cm^2^ in women. Another study performed on patients with CHF aged 73.5 years showed an area of 2.11 cm^2^.

To date, there are few studies that relate muscle contraction to prognostic factors. In our study, we analyzed quadriceps muscle contraction and its relationship with sarcopenia. Marín-Baselga et al. [44] found that a mean contraction of 18% of the quadriceps in patients with HF was associated with more days of hospitalization, sarcopenia measured with the SARC-F test, and frailty. Mateos-Angulo et al. [45] observed a mean contraction of 8.4% in institutionalized patients, related to a high risk of malnutrition. In Table 4, we show that shrinkage below 15% is associated with a higher probability of sarcopenia. We established cut-off points of 15%, 20%, and 30% related to physical function in patients with HF. Further studies expanding the sample size in this group of patients are needed to confirm these findings and close knowledge gaps in this area.

## 5. Limitations

This study has several limitations that should be acknowledged. First, the cross-sectional design of the study limits the ability to establish causality between malnutrition, sarcopenia, and adverse outcomes in heart failure patients. Longitudinal studies are needed to better understand the temporal relationship between these factors.

Second, the sample size, while adequate for preliminary findings, may not be large enough to generalize the results to all patients with heart failure, particularly those at different stages of the disease or those receiving varying treatments. Additionally, the study was conducted in a single center, which may limit the external validity of the findings. Moreover, the performance of anthropometric measurements, dynamometry, and body composition techniques, such as bioimpedancemetry and ultrasound, is necessary to differentiate between men and women due to the inherent differences between the two sexes. This differentiation reduces the sample size.

Third, although we employed comprehensive methods for assessing nutritional status, including the GLIM criteria and bioimpedance analysis, the reliance on clinical assessments and patient self-reports for some data points, such as weight loss and dietary intake, may introduce bias or inaccuracies.

Finally, the study population predominantly consisted of patients in NYHA class I–III, with a smaller representation of those in NYHA class IV. This may impact the applicability of the findings to patients with more severe heart failure, who might exhibit different nutritional and functional profiles.

Future research should aim to address these limitations by incorporating larger, multicenter cohorts and prospective designs to confirm and expand upon the findings presented here.

## 6. Conclusions

Our study found a high prevalence of malnutrition and risk of malnutrition among outpatients with HF, which is associated with worse prognosis. The screening methods evaluated yielded consistent results and proved to be effective in identifying patients at risk. Weight loss was more significant in those who did not survive. In addition, one in four HF patients had a high probability of sarcopenia, and it was higher in patients in NYHA III–IV; furthermore, more than half were considered frail, highlighting the need for adequate morphofunctional assessment in this population. In this assessment, HF patients showed lower PA and inadequate quadriceps muscle contraction, below 30%, which was associated with an increased risk of sarcopenia.

## Figures and Tables

**Figure 1 nutrients-16-02853-f001:**
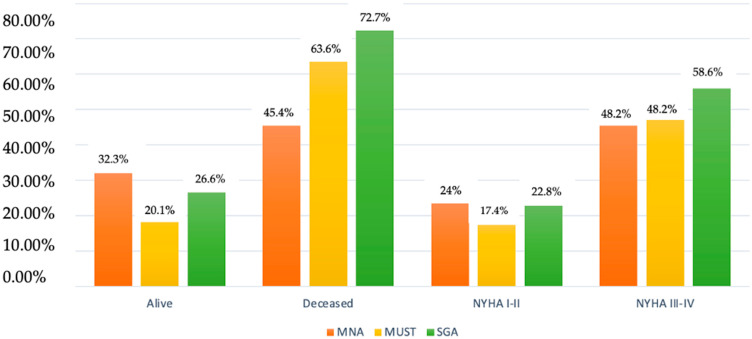
Alive/deceased and NYHA I–IV patients at moderate–high risk of malnutrition according to MNA, MUST, and SGA.

**Table 1 nutrients-16-02853-t001:** Cardiovascular risk factors of patients and distribution by sex.

Cardiovascular Risk Factor	All Patients (*n* = 121)	Males (*n* = 79)	Females (*n* = 42)
HT	79 (65.3%)	49 (62%)	30 (71.4%)
LD	73 (60.3%)	50 (63.3%)	23 (54.8%)
DM type 2	44 (36.4%)	30 (38%)	14 (33.3%)
Obesity	40 (33.1%)	24 (30.4%)	16 (38.1%)

Categorical variables are expressed as percentage (%) of patients. HT = hypertension, LD = dyslipidemia, DM = diabetes.

**Table 2 nutrients-16-02853-t002:** BMI of patients and distribution by sex, in overweight and in obesity.

BMI (kg/m^2^)	All Patients (*n* = 121)	Males (*n* = 79)	Females (*n* = 42)
Total	28.3 ± 5.4	27.7 ± 5	28.2 ± 6.2
Overweight	27.4 ± 1.6	26.9 ± 1.5	27.7 ± 7.6
Obesity	34.6 ± 3.4	33.8 ± 3.3	34.3 ± 4

The variables are expressed as mean ± SD.

**Table 3 nutrients-16-02853-t003:** Diagnosis of malnutrition according to GLIM criteria in all patients, and distribution by sex.

GLIM Criteria	All Patients (*n* = 121)	Males (*n* = 79)	Females (*n* = 42)
No malnutrition	53 (43.8%)	36 (45.5%)	17 (40.4%)
Malnutrition	68 (56.2%)	43 (54.4%)	25 (59.5%)
Moderate malnutrition	55 (87.3%)	37 (86%)	18 (72%)
Severe malnutrition	13 (20.6%)	6 (14%)	7 (28%)

Categorical variables are expressed as percentage (%) of patients.

**Table 4 nutrients-16-02853-t004:** Comparison between malnutrition measured by screening tests (MNA, MUST, and SGA) and malnutrition diagnosed by the GLIM criteria.

Screening Test		GLIM Criteria			*p*-Value
		No Malnutrition	Malnutrition	Severe Malnutrition	
MNA	No malnutrition	25 (75.7%)	19 (65.5%)	0 (0%)	<0.001
	Moderate risk of malnutrition	7 (21.2%)	10 (34.4%)	4 (50%)	
	High risk of malnutrition	1 (3%)	0 (0%)	4 (50%)	
MUST	No malnutrition	47 (88.7%)	44 (80%)	1 (7.6%)	<0.001
	Moderate risk of malnutrition	8 (14.5%)	8 (14.5%)	6 (46.1%)	
	High risk of malnutrition	0 (0%)	3 (5.4%)	6 (46.1%)	
SGA	No malnutrition	43 (81.1%)	39 (70.9%)	1 (7.6%)	<0.001
	Moderate risk of malnutrition	10 (18.8%)	14 (25.4%)	5 (38.4%)	
	High risk of malnutrition	0 (0%)	2 (3.6%)	7 (53.8%)	

Categorical variables are expressed as percentage (%) of patients. MNA = Mini Nutritional Assessment, MUST = Malnutrition Universal Screening Tool, SGA = Subjective Global Assessment.

**Table 5 nutrients-16-02853-t005:** Relationship between nutritional status and SARC-F with NYHA.

	NYHA I–II (*n* = 92)	NYHA III–IV (*n* = 29)	*p*-Value
Glim criteria			0.246
No malnutrition (*n* = 53)	43 (81.1%)	10 (18.8%)	
Malnutrition (*n* = 68)	49 (72%)	19 (27.9%)	
SARC-F			<0.001
Low risk of sarcopenia (*n*= 91)	78 (85.7%)	13 (14.3%)	
High risk of sarcopenia (*n* = 30)	14 (46.6%)	16 (53.4%)	

Categorical variables are expressed as percentage (%) of patients.

**Table 6 nutrients-16-02853-t006:** Description of measurements obtained by BIVA in males.

BIVA Parameter	Mean ± SD	Minimum	Maximum
Total body water (L)	42 ± 9	25.6	64.4
Extracellular water (L)	19.6 ± 3.9	10	31.6
Intracellular water (L)	23.2 ± 4.6	9.6	36.1
Extracellular water ratio	0.4 ± 0	0.3	0.6
Resistance (Ohm)	535.2 ± 88.2	366.3	871.1
Reactance (Ohm)	49.2 ± 11	28	75.3
Phase angle (PA) (°)	5.1 ± 1	3.2	7.2
Lean mass (kg)	49.4 ± 14.6	23.3	80.9
Fat-free mass index (kg/m^2^)	19.2 ± 2.5	11.9	27.3
Musculoskeletal mass index (kg/m^2^)	8.8 ± 1.2	5.8	12.5
Fat mass (kg)	24.2 ± 9.2	6.1	51.6
Fat mass (%)	28.9 ± 7.9	9.5	47
Cell mass (BCM) (kg)	31.6 ± 7	12.6	51.8
ASMI (kg)	22.6 ± 3.5	12.5	32.4
Hydration (%)	73.7 ± 1.9	69.1	83.4

**Table 7 nutrients-16-02853-t007:** Description of measurements obtained by BIVA in females.

BIVA Parameter	Mean ± DS	Minimum	Maximum
Total body water (L)	30.7 ± 4,4	22.9	42.3
Extracellular water (L)	14.7 ± 3.2	8.9	26.7
Intracellular water (L)	15.9 ± 3.2	10.3	25.2
Extracellular water ratio	0.477 ± 0.07	0.376	0.631
Resistance a (Ohm)	620.3 ± 98.6	362	788
Reactance (Ohm)	52.6 ± 12.7	20.6	84.7
Phase angle (PA) (°)	4.7 ± 0.8	2.9	6.4
Lean mass (kg)	35.1 ± 9.5	16.5	48.2
Fat-free mass index (kg/m^2^)	16.7 ± 2.9	8.1	21.4
Musculoskeletal mass index (kg/m^2^)	6.7 ± 1.1	5	10.9
Fat mass (kg)	26.2 ± 11.3	5.4	50.4
Fat mass (%)	36.9 ± 10.3	13.2	51.1
Cell mass (BCM) (kg)	21.8 ± 4.9	13.8	36.1
ASMI (kg)	14.6 ± 2.4	10	20.6
Hydration (%)	73.6 ± 2.8	66.6	89

**Table 8 nutrients-16-02853-t008:** Measurements of the abdominal wall by ultrasound.

	Males	Females
Measurement location (cm)	10 ± 2.2	9 ± 2
Total adipose tissue (mm)	15.5 (9.9–23.5)	20.1 (13.1–27)
Superficial adipose tissue (mm)	6.7 (4.2–9.2)	9 (5.2–14.7)
Preperitoneal adipose tissue (mm)	4.9 (3.4–7.9)	5 (3.4–5.9)

The variables are expressed as mean ± SD or median (IQR).

**Table 9 nutrients-16-02853-t009:** Measurements of the anterior rectus quadriceps muscle by ultrasound.

	Males	Females
Measurement location (cm)	16.8 ± 1.3	16.3 ± 1.4
X axis (mm)	32.1 (27.2–35.5)	30.6 (24.7–33.9)
Y axis (mm)	9 (7–12.8)	9 (7–11)
Area (cm^2^)	2.4 ± 1.5	1.9 ± 1.1
Adipose tissue (mm)	5.5 (3.8–7.6)	11.6 (8.1–17.4)

The variables are expressed as mean ± SD or median (IQR).

**Table 10 nutrients-16-02853-t010:** Sarcopenia according to SARC-F with respect to contraction of the anterior rectus quadriceps muscle.

		Low Risk of Sarcopenia (SARC-F)	High Risk of Sarcopenia (SARC-F)	*p*-Value
Contraction >15%	Yes (*n* = 38)	33 (86.8%)	5 (13.1%)	0.045
	No (*n* = 51)	35 (68.6%)	16 (31.3%)	
Contraction >20%	Yes (*n* = 33)	29 (87.8%)	4 (12.1%)	0.050
	No (*n* = 56)	39 (69.6%)	17 (30.3%)	
Contraction >30%	Yes (*n* = 21)	20 (95.2%)	1 (4.7%)	0.020
	No (*n* = 68)	48 (70.5%)	20 (29.4%)	

Categorical variables are expressed as percentage (%) of patients. SARC-F = Strength, Assistance in walking, getting up from a chair, climbing stairs, falls.

## Data Availability

The data presented in this study are available on request from the corresponding author due to time limitations.

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
