# Peer review of "Nutritional Assessment in Outpatients with Heart Failure"

_nutrients, 2024, doi:10.3390/nu16172853_

Round 1
Reviewer 1 Report
Comments and Suggestions for Authors
Guillén et al determine the prevalence of malnutrition and risk of malnutrition in outpatients with HF. However, the topic is not new.
1) MNA has been evaluated in only some patients.
2) This study was a cross-sectional analysis, small sample size, and the lack of repetitive malnutrition assessments.
3) The authors have conducted many evaluations, but the results are not well integrated and the text is redundant.
4) The readers may be interested in the association of cachexia and sarcopenia with clinical outcomes.
Comments on the Quality of English Language
I recommend that you have this paper proofread in English.
Author Response
For research article: “Nutritional assessment in outpatients with heart failure”
|
Response to Reviewer 1 Comments
|
|||
|
1. Summary |
|
|
|
|
Thank you very much for taking the time to review this manuscript. Please find the detailed responses below and the corresponding revisions highlighted changes in the re-submitted files.
|
|||
|
2. Questions for General Evaluation |
Reviewer’s Evaluation |
Response and Revisions |
|
|
Does the introduction provide sufficient background and include all relevant references? |
Must be improved |
After revising the introduction, I have expanded the section on the pathophysiology of malnutrition in heart failure to highlight the connection between the inflammatory state associated with HF and the development of malnutrition, emphasizing the importance of its prevention. This is crucial, as malnutrition may be present subclinically before it is diagnosed. In addition, I have incorporated more references that enrich the content and provide a stronger context to this part. |
|
|
|
|
||
|
Is the research design appropriate? |
Must be improved
|
We have done a cross-sectional study; I agree that it could be interesting to conduct a prospective study. In this sense, I have added a "limitations" section (line 453, page 12) in which I explain that our idea is that our work serves as a basis for future prospective studies on malnutrition in HF.
|
|
|
Are the methods adequately described? |
Can be improved
|
Agree, accordingly we have added in the study design section, in the line 124, paragraph 3, page 3 that the patient selection process was randomized from cardiology outpatient clinics.
|
|
|
Are the results clearly presented? |
Can be improved
|
We agree. We have added three tables in the section "patient characteristics and nutritional assessment" on page 5 and 6 to make it more visual and easier to read. As well, we have added a diagram (page 6) representing the screening for malnutrition according to MNA, MUST and SGA in deceased patients and in those in NYHA III and IV.
|
|
|
Are the conclusions supported by the results? |
Can be improved
|
Agree. We have, accordingly, done this modification.
|
|
|
3. Point-by-point response to Comments and Suggestions for Authors |
|||
|
Comments 1: MNA has been evaluated in only some patients.
|
|||
|
Response 1: I agree with this comment. Indeed, the nutrition screening assessment with the MNA test was performed only on those patients who were older than 65 years at the time of data collection. The reason is that this is the population in which this test is validated. Below is the bibliography of the validation (Spanish version) of the MNA test in patients over 65 years of age.
Muñoz Díaz B, Molina-Recio G, Romero-Saldaña M, Redondo Sánchez J, Aguado Taberné C, Arias Blanco C, Molina-Luque R, Martínez De La Iglesia J. Validation (in Spanish) of the Mini Nutritional Assessment survey to assess the nutritional status of patients over 65 years of age. Fam Pract. 2019 Mar 20;36(2):172-178. doi: 10.1093/fampra/cmy051. Erratum in: Fam Pract. 2019 Jul 31;36(4):528. doi: 10.1093/fampra/cmy093. PMID: 29873713.
|
|||
|
Comments 2: This study was a cross-sectional analysis, small sample size, and the lack of repetitive malnutrition assessments.
|
|||
|
Response 2: Thank you for pointing this out. I agree. Therefore, I have included a “Limitations” section which can be found on line 453, page 12. While it is true that the study´s cross-sectional nature is a limitation, we believe it effectively underscores the importance of malnutrition in heart failure and can serve as a basis for future research in this area.
Comments 3: The authors have conducted many evaluations, but the results are not well integrated and the text is redundant.
Response 3: Thank you for pointing this out. To better integrate the results and make them clearer and easier to read we have added several tables in the first results section "patient characteristics and nutritional assessment". The first table shows the cardiovascular risk factors in the entire sample and the distribution by sex. The second table shows the BMI of the total sample, in those with overweight and obesity, also in the total population and the distribution by sex. In the third one we have represented the diagnosis of malnutrition according to GLIM criteria. In addition to all the above, we have added a diagram representing the screening for malnutrition according to MNA, MUST and SGA in deceased patients and in those in NYHA III and IV.
Comments 4: The readers may be interested in the association of cachexia and sarcopenia with clinical outcomes.
Response 4: I totally agree with this comment. In the introduction section, starting on line 92, paragraph 5, page 2 we have named three studies that show how those patients with sarcopenia and cachexia have worse prognosis in terms of survival and comorbidities. However, our study does not aim to find a relationship between malnutrition and prognostic factors and this is a limitation that we have left explained in line 454, paragraph 1, page 12. Nevertheless, we believe that our study is very useful as it demonstrates the prevalence of malnutrition and sarcopenia in these patients and serves as a basis for future prospective studies in patients with HF.
|
|||
|
4. Response to Comments on the Quality of English Language |
|||
|
Point 1: Quality of English Language (x) Moderate editing of English language required. |
|||
|
Response 1: The English used for the article has been reviewed by a native professional prior to submission. If there are any particular expressions or words that need to be modified or changed, please let me know. However, it will be revised again.
|
|||

Reviewer 2 Report
Comments and Suggestions for Authors
Dear authors,
I have read your paper entitled “Nutritional assessment in outpatients with heart failure”. The topic is interesting, but there are a few things to add:
1. Please rewrite the abstract section to provide an introduction, material and methods, results, and conclusion sections.
2. Please provide a diagram describing the patient selection process for the study.
3. Please insert a table containing all general data of the patients enrolled in the study.
4. It is necessary to add a sub-chapter discussing the types of diets that patients included in the study should adopt.
5. Please add a paragraph describing the limitations.
Author Response
For research article: ““Nutritional assessment in outpatients with heart failure”
|
Response to Reviewer 2 Comments
|
|||
|
1. Summary |
|
|
|
|
Thank you very much for taking the time to review this manuscript. Please find the detailed responses below and the corresponding revisions/corrections highlighted/in track changes in the re-submitted files.
|
|||
|
2. Questions for General Evaluation |
Reviewer’s Evaluation |
Response and Revisions |
|
|
Does the introduction provide sufficient background and include all relevant references? |
Must be improved |
After revising the introduction, we have expanded the section on the pathophysiology of malnutrition in heart failure to highlight the connection between the inflammatory state associated with HF and the development of malnutrition, emphasizing the importance of its prevention. This is crucial, as malnutrition may be present subclinically before it is diagnosed. In addition, I have incorporated more references that enrich the content and provide a stronger context to this part.
|
|
|
|
|
||
|
Is the research design appropriate? |
Can be improved |
Thank you for pointing this out. I agree. Therefore, we have included a “Limitations” section which can be found on line 454, page 12. While it is true that the study´s cross-sectional nature is a limitation, we believe it effectively underscores the importance of malnutrition in heart failure and can serve as a basis for future research in this area. |
|
|
Are the methods adequately described? |
Can be improved |
Agree, accordingly we have added in the study design section, in the line 124, paragraph 3, page 3 that the patient selection process was randomized from cardiology outpatient clinics.
|
|
|
Are the results clearly presented? |
Must be improved |
We agree. We have added three tables in the section "patient characteristics and nutritional assessment" on page 5 and 6 to make it more visual and easier to read. As well, we have added a diagram (page 6) representing the screening for malnutrition according to MNA, MUST and SGA in deceased patients and in those in NYHA III and IV.
|
|
|
Are the conclusions supported by the results? |
Can be improved |
Agree. We have, accordingly, done this modification.
|
|
|
3. Point-by-point response to Comments and Suggestions for Authors
|
|||
|
Comments 1: Please rewrite the abstract section to provide an introduction, material and methods, results, and conclusion sections.
Response 1: Agree. We have, accordingly, done this modification. |
|||
|
|
|||
|
Comments 2: Please provide a diagram describing the patient selection process for the study.
Response 2: Thank you for pointing this out. The patient selection process was randomized on the basis of the cardiology outpatient clinic outpatient agenda. However, we have not included the diagram with the selection process since we simply chose randomly from patients between 18 and 80 years of age who were under active treatment for heart failure. We called them by telephone and explained the purpose of the study and what it was going to consist of. If they accepted, they attended and were given the documentation described in the "material and methods" section: the information sheet and the informed consent form. We have added in the "study design" section on line 124, paragraph 3, page 3 this detail: "Patients were randomly selected from the monographic consultation of heart failure in cardiology outpatient clinics”.
Comments 3: Please insert a table containing all general data of the patients enrolled in the study.
Response 3: Agree. We have, revised the "patients characteristics and nutritional assessment" section and we have added a table with the cardiovascular risk factors in the total population included in the study and the distribution by sex. Likewise, we have added a table with the total BMI, in those with overweight and obesity, in the total population and divided by sex. We have also added a table showing the diagnosis of malnutrition according to GLIM criteria in the total population and divided by sex. We have done this in three different tables since some variables are categorical and others numerical. We agree that this way it is clearer.
Comments 4: It is necessary to add a sub-chapter discussing the types of diets that patients included in the study should adopt.
Response 4: Thank you for your comment. However, we disagree on this issue. We have not introduced a subchapter about the different diets that should be followed by the patients included in the study since this was not the objective of the study. As they are patients with heart failure and present pulmonary and peripheral congestion, the diet they should generally adopt should be low in salt and heart-healthy. In other words, they should avoid ultra-processed foods, industrial pastries and fried foods and prioritize healthy fats and protein, both animal and vegetable.
Comments 5: Please add a paragraph describing the limitations.
Response 1: Agree. We have, accordingly, done this modification. |
|||
|
|
|||
|
4. Response to Comments on the Quality of English Language |
|||
|
Point 1: (x) English language fine. No issues detected. |
|||
|
Response 1: Agree
|
|||

Reviewer 3 Report
Comments and Suggestions for Authors
To the Authors
The original article, entitled Nutritional assessment in outpatients with heart failure aims to determine the prevalence of malnutrition and risk of malnutrition in outpatients with HF. After carefully following the submitted article, I made the following findings:
The introduction section should bring more results of previous studies to justify the aim of the research:
- Line 61-62 Prevention and detection of malnutrition in HF patients is essential to improving their prognosis. This is a very important aspect which deserves more extensive explanation. Previous results on this point should be presented to provide an adequate background of the current knowledge on the role of malnutrition in patients with heart failure.
- Line 66-67 The main objective of the study was to determine the prevalence of malnutrition and risk of malnutrition in outpatients with HF, using the GLIM criteria and morphofunctional assessment [14]. The authors should present more clearly the motivation for choosing this study objective.
- The objective of the study is not clearly stated – the authors compared frequency data provided by different screening instruments and functional tests that evaluate malnutrition. The current design of the study is not suitable for the purpose of the study.
Material and methods
In the results section, results from the deceased patients are presented. The inclusion of deceased persons means a follow-up of patients over time. From this point of view, the design of the study is not an observational one, but a prospective or retrospective one, and this must be specified in the material and methods. The authors must specify the number of patients who died among the 122 included in the analysis.
Line 100 the words ‘In turn’ are not necessary.
Line 151-152 - The SARC-F test was provided by Parra-Rodriguez et al. in The Journal of Post-151 Acute and Long-Term Care Medicine, volume 17, issue 12, available at 152 (https://www.jamda.com/) – this is a citation – it should be included in the reference. If special permission was requested for the use of this method, it should be included in the Acknowledgments section,
Results
Line 193 - the should be removed
Line 195-197 - Among the deceased patients, 45.4% were 195 at moderate-to-high risk of malnutrition according to MNA; 63.6% according to MUST 196 and 72.7% according to SGA – the author did not previously explain the inclusion of the deceased patients in the analysis. The number of patients who died among the 122 should be presented in the analysis.
Table 1 –( Table 1. Comparison between malnutrition measured by screening tests (MNA, MUST and SGA) and malnutrition diagnosed by the GLIM criteria.) presents the result of Chi Square test. A kappa-Cohen agreement test would better serve the purpose of the study because it considers the possibility of the agreement occurring by chance. This observation is also applicable to the results presented in Table 6.
Author Response
For research article: ““Nutritional assessment in outpatients with heart failure”
|
Response to Reviewer 3 Comments
|
|||
|
1. Summary |
|
|
|
|
Thank you very much for taking the time to review this manuscript. Please find the detailed responses below and the corresponding revisions/corrections highlighted/in track changes in the re-submitted files.
|
|||
|
2. Questions for General Evaluation |
Reviewer’s Evaluation |
Response and Revisions |
|
|
Does the introduction provide sufficient background and include all relevant references? |
Must be improved |
After revising the introduction, I have expanded the section on the pathophysiology of malnutrition in heart failure to highlight the connection between the inflammatory state associated with HF and the development of malnutrition, emphasizing the importance of its prevention. This is crucial, as malnutrition may be present subclinically before it is diagnosed. In addition, I have incorporated more references that enrich the content and provide a stronger context to this part.
|
|
|
|
|
||
|
Is the research design appropriate? |
Must be improved |
We have done a cross-sectional study; I agree that it could be interesting to conduct a prospective study. In this sense, I have added a "limitations" section (line 453, page 12) in which I explain that our idea is that our work serves as a basis for future prospective studies on malnutrition in HF.
|
|
|
Are the methods adequately described? |
Yes |
|
|
|
Are the results clearly presented? |
Must be improved |
We agree. We have added three tables in the section "patient characteristics and nutritional assessment" on page 5 and 6 to make it more visual and easier to read. As well, we have added a diagram (page 6) representing the screening for malnutrition according to MNA, MUST and SGA in deceased patients and in those in NYHA III and IV.
|
|
|
Are the conclusions supported by the results? |
Must be improved |
Agree. We have, accordingly, done this modification.
|
|
|
3. Point-by-point response to Comments and Suggestions for Authors
|
|||
|
Comments 1: Prevention and detection of malnutrition in HF patients is essential to improving their prognosis. This is a very important aspect which deserves more extensive explanation. Previous results on this point should be presented to provide an adequate background of the current knowledge on the role of malnutrition in patients with heart failure.
Response 1: Agree. In accordance with the comment, we have reviewed existing literature on studies dealing with the prevalence of malnutrition in HF. Several studies have found a high prevalence of malnutrition in these patients and that it is related to worse prognostic factors. All this justifies the need for adequate prevention, and we have added the following (line 90, paragraph 5, page 2).
“Prevention and detection of malnutrition in HF patients is essential to improve their prognosis. In this regard, regardless of the method of nutritional assessment used, several studies have shown that malnutrition is an independent predictor of mortality. Lv, Ru et al. wrote a meta-analysis [10] that included 31 studies (12,537 patients with HF), it was shown that mortality in malnourished patients was twice as high as in normonnourished patients regardless of age. Jiang et al. conducted a study [19] that enrolled 209 patients hospitalized in the Intensive Care Unit with severe cardiovascular disease; 92.34% were at risk of malnutrition and 18.18% were at high risk of malnutrition. In their sample, the higher the nutritional risk, the higher the incidence of cardiovascular events and the higher the mortality. Guerra Sánchez et al. [20] included 76 patients hospitalized with HF; 22.4% were malnourished and 77.6% were at risk of malnutrition according to SGA. In the group that underwent nutritional intervention, analytical parameters improved (increase in hemoglobin, total protein and albumin) and the results in the 6-minute walk test. Evaluating the nutritional status of these patients is crucial as it allow guiding the nutritional approach necessary to prevent or reverse malnutrition, contributing to improve their prognosis.”
|
|||
|
|
|||
|
Comments 2: The main objective of the study was to determine the prevalence of malnutrition and risk of malnutrition in outpatients with HF, using the GLIM criteria and morphofunctional assessment [14]. The authors should present more clearly the motivation for choosing this study objective. The objective of the study is not clearly stated – the authors compared frequency data provided by different screening instruments and functional tests that evaluate malnutrition. The current design of the study is not suitable for the purpose of the study.
Response 2: Thank you for pointing this out. In agreement with the comment, we have added the following to the objective to give more background and understanding of the objective: As we have seen, malnutrition in patients with HF is a poor prognostic factor that compromises the patient's health and can lead to death, as in the case of cardiac cachexia. Therefore, the assessment of nutritional status in patients with HF is essential and will allow us to guide the nutritional approach to reverse the situation that compromises the patient's life. Before being able to carry out nutritional intervention studies on these patients, it is necessary to know the reality of the nutritional status of outpatients with HF, hence, the main objective of the study was to determine the prevalence of malnutrition and risk of malnutrition in outpatients with HF, using the GLIM criteria and morphofunctional assessment.
Likewise, we have changed the wording of the secondary objective as we seek to see if nutritional screening tests are related to the diagnosis of malnutrition according to GLIM criteria, in order to see if there is any screening test that is more useful in the detection of malnutrition in these patients: As a secondary objective, the relationship between different nutritional screening tests and the GLIM criteria in HF patients was tested.
Comments 3: In the results section, results from the deceased patients are presented. The inclusion of deceased persons means a follow-up of patients over time. From this point of view, the design of the study is not an observational one, but a prospective or retrospective one, and this must be specified in the material and methods. The authors must specify the number of patients who died among the 122 included in the analysis.
Response 3: Thank you for the comment. The study we present is observational and cross-sectional because we evaluated the patients only once and performed all the screening tests and the nutritional assessment. However, the recruitment of the patients lasted two years and during that time, 11 died, is included in the line 228, paragraph 4, page 5): 11 (9.1%) died during data collection, of which 6 were male and 5 female.
Comments 4: The SARC-F test was provided by Parra-Rodriguez et al. in The Journal of Post-151 Acute and Long-Term Care Medicine, volume 17, issue 12, available at 152 (https://www.jamda.com/) – this is a citation – it should be included in the reference. If special permission was requested for the use of this method, it should be included in the Acknowledgments section.
Response 4: Agree. We have, accordingly, done this modification.
Comments 5: Among the deceased patients, 45.4% were 195 at moderate-to-high risk of malnutrition according to MNA; 63.6% according to MUST 196 and 72.7% according to SGA – the author did not previously explain the inclusion of the deceased patients in the analysis. The number of patients who died among the 122 should be presented in the analysis.
Response 5: The number of deceased patients is on the line 228 under "Patient Characteristics and Nutritional Assessment" in the results section.
Comments 6: Table 1. Comparison between malnutrition measured by screening tests (MNA, MUST and SGA) and malnutrition diagnosed by the GLIM criteria.) presents the result of Chi Square test. A kappa-Cohen agreement test would better serve the purpose of the study because it considers the possibility of the agreement occurring by chance. This observation is also applicable to the results presented in Table 6.
Response 6: Agree. The statistics part of our article was reviewed by a statistics team. We thought about whether to do Kappa-cohen or Chi-square on both tables. Finally, we decided to perform the analysis with Chi-square since the objective is not to look for correlation but to see, in Table 1 (which with the modifications is now Table 4) if the screening tests evaluated are valid to correctly detect malnutrition in these patients and therefore to see the relationship in those who are normonnourished and malnourished with the GLIM criteria. Similarly in Table 6 (now Table 9) the aim was to determine how the quadriceps muscle contraction was in patients with and without sarcopenia according to SARC-F. Again, we were not looking to see if there was a correlation as such, we simply wanted to study the relationship between the two variables. |
|||
|
|
|||
|
4. Response to Comments on the Quality of English Language |
|||
|
Point 1: (x) English language fine. No issues detected. |
|||
|
Response 1: Agree
|
|||
Reviewer 4 Report
Comments and Suggestions for Authors
This manuscript describes the results of nutritional assessment of outpatients with heart failure using various screening tools and the GLIM criteria, as well as the risk of sarcopenia. A detailed investigation was conducted and the report is very well written. I have few comments to the author.
Comment 1: In this study, measurements were performed using ultrasound, but how common is this measurement method? If it is a common method, I think it would be better to show the references.
Comment 2: In the "Discussion", past reports with description of the impact of the proportion of NYHA III-IV patients, were coted. However, in this study, the results of nutrition screening/assessment and Sarc-F according to NYHA classification are not shown in the table. Since NYHA status is thought to affect the results of nutrition screening/assessment and Sarc-F, I think it would be better to indicate NYHA status snd nutrition screening/assessment/Sarc-F as a table.
Author Response
For research article: ““Nutritional assessment in outpatients with heart failure”
|
Response to Reviewer 4 Comments
|
|||||||||||||||||||||||||||
|
1. Summary |
|
|
|||||||||||||||||||||||||
|
Thank you very much for taking the time to review this manuscript. Please find the detailed responses below and the corresponding revisions/corrections highlighted/in track changes in the re-submitted files.
|
|||||||||||||||||||||||||||
|
2. Questions for General Evaluation |
Reviewer’s Evaluation |
Response and Revisions |
|||||||||||||||||||||||||
|
Does the introduction provide sufficient background and include all relevant references? |
Yes |
|
|||||||||||||||||||||||||
|
|
|
||||||||||||||||||||||||||
|
Is the research design appropriate? |
Yes |
|
|||||||||||||||||||||||||
|
Are the methods adequately described? |
Can be improved |
Agree, accordingly we have added in the study design section, in the line 124, paragraph 3, page 3 that the patient selection process was randomized from cardiology outpatient clinics.
|
|||||||||||||||||||||||||
|
Are the results clearly presented? |
Can be improved |
We agree. We have added three tables in the section "patient characteristics and nutritional assessment" on page 5 and 6 to make it more visual and easier to read. As well, we have added a diagram (page 6) representing the screening for malnutrition according to MNA, MUST and SGA in deceased patients and in those in NYHA III and IV.
|
|||||||||||||||||||||||||
|
Are the conclusions supported by the results? |
Yes |
|
|||||||||||||||||||||||||
|
3. Point-by-point response to Comments and Suggestions for Authors
|
|||||||||||||||||||||||||||
|
Comments 1: In this study, measurements were performed using ultrasound, but how common is this measurement method? If it is a common method, I think it would be better to show the references.
Response 1: Thank you for pointing this out. We agree, nutritional ultrasound is an emerging technique that has gained attention in recent years for its potential in assessing muscle mass and adipose tissue in clinical nutrition. However, it is important to acknowledge that this method is not yet widely implemented across all nutrition services, and standardized reference values are still under development. While its use is currently more common in research settings and specialized clinics, we believe that its inclusion adds value to our study by providing more detailed and specific measurements of muscle and adipose tissue, which are crucial for the accurate assessment of sarcopenia and malnutrition.
Accordingly, we have added in the discussion a note about it in the line 422, paragraph 3, page 11: “Regarding nutritional ultrasound, in this study, it was utilized to assess muscle mass and adipose tissue as part of the morphofunctional evaluation of patients with heart failure. It is important to note that while nutritional ultrasound is an emerging tool with significant potential for enhancing the accuracy of body composition assessments, it is not yet widely adopted in all clinical nutrition services. The technique is currently more common in research settings and specialized clinics, and the establishment of standardized reference values is still in progress. Despite these limitations, the inclusion of ultrasound in this study provided valuable insights into the muscle and adipose tissue characteristics of our patient population, which are critical for the accurate diagnosis of sarcopenia and malnutrition.” |
|||||||||||||||||||||||||||
|
|
|||||||||||||||||||||||||||
|
Comments 2: In the "Discussion", past reports with description of the impact of the proportion of NYHA III-IV patients, were coted. However, in this study, the results of nutrition screening/assessment and Sarc-F according to NYHA classification are not shown in the table. Since NYHA status is thought to affect the results of nutrition screening/assessment and Sarc-F, I think it would be better to indicate NYHA status and nutrition screening/assessment/Sarc-F as a table.
Response 2: I fully agree with that statement. Consequently, we have added a table (table 5) on page 7 which is copied below.
|
|||||||||||||||||||||||||||
|
|
|||||||||||||||||||||||||||
|
4. Response to Comments on the Quality of English Language |
|||||||||||||||||||||||||||
|
Point 1: (x) English language fine. No issues detected. |
|||||||||||||||||||||||||||
|
Response 1: Agree
|
|||||||||||||||||||||||||||
Round 2
Reviewer 1 Report
Comments and Suggestions for Authors
Authors et al determined the malnutrition in HF. However, the issue is not new and this paper has some serious problems.
Major limitation
1) Duration of heart failure is unknown. If HF is an event that occurred many years ago, HF will affect malnutrition. This is the most serious limitation.
2) This study was a cross-sectional analysis, and small sample size.
3) Please show patient characteristics (age, sex, BMI, LVEF, …).
4) The authors do not compare against other nutritional assessments.
5) Introduction and discussion are redundant.
6) Figure 1 shows survival vs. death, NYHA 1-2 vs. 3-4.
7) The study should be reorganized and presented again as to what it is that this study is intended to show.
Author Response
|
1. Summary |
|
|
|
|
Thank you very much for taking the time to review this manuscript. Please find the detailed responses below and the corresponding corrections in track changes in the re-submitted files.
|
|||
|
2. Questions for General Evaluation
|
Reviewer’s Evaluation |
Response and Revisions |
|
|
Does the introduction provide sufficient background and include all relevant references?
|
Must be improved |
Thank you very much for highlighting this. We agree that the part about the pathophysiology of malnutrition in HF may be redundant. Therefore, we have summarized the introductory section to make it easier to read. |
|
|
|
|
||
|
Is the research design appropriate?
|
Must be improved
|
We agree in this comment. Hence, in the “limitations” section on line 453, paragraph 2, page 12, we have included the low sample size and the fact that it is single-center since it may be an impediment to generalizing the results obtained. However, we strongly believe that it can serve as a basis for multicenter studies and for expanding the sample size for future research. Likewise, it can also be the basis for studies that seek causality between malnutrition and poor prognosis in these patients.
|
|
|
Are the methods adequately described?
|
Can be improved
|
Agree, accordingly we have added in the study design section, in the line 124, paragraph 3, page 3 that the patient selection process was randomized from cardiology outpatient clinics.
|
|
|
Are the results clearly presented?
|
Must be improved
|
Agreed. We have added the heart failure evolution time of our sample in the line 229, paragraph 4, page 5:” The duration of HF was 6 (3-11) years”. As well, we added in Figure 1 the patients who were alive and those in NYHA I-II so that the comparison with the deceased and NYHA III-IV in terms of risk of malnutrition could be seen. The figure can be seen in line 255, paragraph 3, page 6. |
|
|
Are the conclusions supported by the results? |
Must be improved
|
Agree. We have, accordingly, done this modification.
|
|
|
3. Point-by-point response to Comments and Suggestions for Authors |
|||
|
Comments 1: Duration of heart failure is unknown. If HF is an event that occurred many years ago, HF will affect malnutrition. This is the most serious limitation.
Response 1: Thank you for pointing this out. We agree in this comment. Therefore, we have added the heart failure evolution time of our sample in the line 229, paragraph 4, page 5:” The duration of HF was 6 (3-11) years”. As this was a cross-sectional study, our aim was to analyze a group of patients with HF at the time of data collection in an attempt to make it representative of the general population. Our study places special emphasis on the importance of preventing HF malnutrition. It is correct that the more years of HF evolution, the greater the probability of malnutrition if the correct approach has not been taken, given that it produces chronic inflammation. However, by collecting a heterogeneous sample of patients in terms of the evolution of HF, we can see the real impact of malnutrition regardless of the time of evolution. Furthermore, our study does not aim to objectify causality, but rather to be a descriptive study of the nutritional data of these patients in order to be able to carry out an optimal nutritional approach. |
|||
|
|
|||
|
Comments 2: This study was a cross-sectional analysis, and small sample size.
Response 2: We agree in this comment. Hence, in the “limitations” section on line 453, paragraph 2, page 12, we have included the low sample size and the fact that it is single-center since it may be an impediment to generalizing the results obtained. However, we strongly believe that it can serve as a basis for multicenter studies and for expanding the sample size for future research. Likewise, it can also be the basis for studies that seek causality between malnutrition and poor prognosis in these patients. |
|||
|
Comments 3: Please show patient characteristics (age, sex, BMI, LVEF, …). Response 3: Thank you for this comment, as it is of vital importance. The patient characteristics are found in the “results” section in the subsection “patients characteristics and nutritional assessment” on line 227, paragraph 3, page 5. In written form are the characteristics in terms of age, sex, LVEF and NYHA scale. Tables 1 and 2 show the cardiovascular risk factors and BMI in total, overweight and obesity.
Comments 4: The authors do not compare against other nutritional assessments. Response 4: Thank you for pointing this out. We agree that we could have compared more nutritional screening tests but choosing these three makes sense. The MNA is quick to perform and assesses weight loss, intake and anthropometric measurements. The VGS is independent of the concentration of biochemical parameters and assesses weight loss, reduced intake, body fat and edema. Finally, the MUST test, recommended by the British Association of Parenteral and Enteral Nutrition due to its speed and simplicity, was used. There are several studies that support its use in patients with HF, the references of which are given below: - Guerra-Sánchez, Luis, Martínez-Rincón, Carmen, & Fresno-Flores, Mar. (2015). Cribado nutricional en pacientes con insuficiencia cardiaca: análisis de 5 métodos. Nutrición Hospitalaria, 31(2), 890-899. doi: 10.3305/nh.2015.31.2.7665 - Guerra-Sánchez L, Fresno-Flores M, Martínez-Rincón C. Effect of a double nutritional intervention on the nutritional status, functional capacity, and quality of life of patients with chronic heart failure: 12-month results from a randomized clinical trial. Nutr Hosp [Internet]. 2020 [citado 19 de mayo de 2024]; Disponible en: https://www.nutricionhospitalaria.org/articles/02820/show - Barbosa JS, Souza MFC, Costa JO, Alves LVS, Oliveira LMSM, Almeida RR, Oliveira VB, Pereira LMC, Rocha RMS, Costa IMNBC, Vieira DADS, Baumworcel L, Almeida-Santos MA, Oliveira JLM, Neves EB, Díaz-de-Durana AL, Merino-Fernández M, Aidar FJ, Sousa ACS. Assessment of Malnutrition in Heart Failure and Its Relationship with Clinical Problems in Brazilian Health Services. Int J Environ Res Public Health. 2022 Aug 15;19(16):10090. doi: 10.3390/ijerph191610090. PMID: 36011722; PMCID: PMC9408367. - Duarte RRP, Gonzalez MC, Oliveira JF, Goulart MR, Castro I. Is there an association between the nutritional and functional parameters and congestive heart failure severity? Clin Nutr. mayo de 2021;40(5):3354-9. - Joaquín C, Puig R, Gastelurrutia P, Lupón J, De Antonio M, Domingo M, et al. Mini nutritional assessment is a better predictor of mortality than subjective global assessment in heart failure out-patients. Clin Nutr. diciembre de 2019;38(6):2740-6. - Hu Y, Yang H, Zhou Y, Liu X, Zou C, Ji S, Liang T. Prediction of all-cause mortality with malnutrition assessed by nutritional screening and assessment tools in patients with heart failure:a systematic review. Nutr Metab Cardiovasc Dis. 2022 Jun;32(6):1361-1374. doi: 10.1016/j.numecd.2022.03.009. Epub 2022 Mar 14. PMID: 35346547.
Given that these three tests have been shown to be useful in the screening for malnutrition in HF, we thought it appropriate to limit ourselves to their performance in order to clarify which is the best test for screening in these patients.
Comments 5: Introduction and discussion are redundant. Response 5: Thank you very much for highlighting this. We agree that the part about the pathophysiology of malnutrition in HF may be redundant. Therefore, we have summarized the introductory section to make it easier to read.
Comments 6: Figure 1 shows survival vs. death, NYHA 1-2 vs. 3-4. Response 6: Agreed. We have added in Figure 1 the patients who were alive and those in NYHA I-II so that the comparison with the deceased and NYHA III-IV in terms of risk of malnutrition could be seen. The figure can be seen in line 255, paragraph 3, page 6.
|
|||
|
4. Response to Comments on the Quality of English Language |
|||
|
Point 1: I am not qualified to assess the quality of English in this paper. |
|||
|
|
|||

Reviewer 2 Report
Comments and Suggestions for Authors
The authors did not fulfill requirement number 4.
Author Response
Thank you very much for taking the time to review the article we have prepared. I will now proceed to respond to the comments arising from the review.
Comment 1: The authors did not fulfill requirement number 4.
Response 1: Thank you for your comment. We have added on line 121, paragraph 3, page 3 the type of diet the patients were on: “The entire sample was on a low-salt diet according to the recommendations of their cardiology physicians.” However, as commented in the previous review, we have not added a subsection with the type of diet that the included patients should adopt since this is a retrospective study in which we have not performed nutritional intervention. Nevertheless, our study may serve as a basis for future prospective studies with the aim of nutritional therapy.
Reviewer 3 Report
Comments and Suggestions for Authors
Dear authors,
I am pleased to notice that your manuscript has significantly improved in quality with the newest version.
However, I would kindly ask of you read through it several times more, as I have noticed several spelling and formatting errors.
Author Response
Thank you very much for taking the time to review the article we have prepared. I will now proceed to respond to the comments arising from the review.
Comment 1: I would kindly ask of you read through it several times more, as I have noticed several spelling and formatting errors.
Response 1: Thank you very much for your appreciation. We agree and will review the spelling and grammar of the article in order to improve.